# ProCEED: Prototype Consolidation and Ensemble-based Exemplar-Free Deep Incremental Learning

## Abstract

Exemplar-free Class Incremental Learning requires the learning agent to incrementally acquire new class information and maintain past knowledge without having access to samples from previous tasks. Despite the significant performance achieved by the subspace ensemble of a mixture of experts (MoE) with Gaussian prototypical networks, a critical gap still exists. As the downstream tasks arrive, the subspace representation of old classes gets updated, resulting in a prototype drift and leading to forgetting. To address the forgetting problem, we propose ProCEED to dynamically realign previous classes' representation in the latest subspace to adjust the drifted class prototypes and preserve their decision boundaries. Specifically, we compute the inter-subspace angular drifts of the prototype of previous incremental stages with the current one, holding the local semantic relationship between the incremental subspaces. The angular drift is then used to adjust old tasks' prototypes into the subspace of incremental tasks. Furthermore, the model inherits combined knowledge from MoE, supporting plasticity without extra computational burden. Consequently, ProCEED significantly balances the stability-plasticity dilemma over incoming incremental tasks, allowing the model to learn continually. The experimental evaluations on challenging benchmark datasets demonstrate dominant accuracy for ProCEED compared to the state-of-the-art class-incremental learning methods.

## 1 Introduction

Intelligent learning machines should imitate human learning ability to accumulate knowledge while adapting to dynamically changing environments without the availability of previous information. However, when a traditional learning model is designed to learn a sequence of tasks from streaming datasets, the previously learned parameters are overwritten by the current task, and the model suffers from *Catastrophic Forgetting* McCloskey & Cohen (1989); French (1999); Kirkpatrick et al. (2017); Li & Hoiem (2017); Lopez-Paz & Ranzato (2017); Schwarz et al. (2018); Zenke et al. (2017). To address this issue of forgetting, in recent years, the deep learning community has shifted its attention towards Class Incremental Learning (CIL), where the primary goal is to learn to classify all previously seen classes from sequences of tasks Zhu et al. (2022). The forgetting problem can be naively resolved by rehearsing representative samples from previous tasks. However, rehearsing samples increases the computational cost linearly with tasks and raises data privacy questions due to the requirement of continuous access to sensitive data, especially in the medical sector and national security Goswami et al. (2024).

Exemplar-free class-incremental Learning (EFCIL), a sub-field of CIL, is a challenging learning paradigm that seeks to mitigate forgetting without storing samples from previous tasks. In the literature, researchers tend to use 50% of the training samples upfront to learn a strong feature extractor and freeze it after the first task (known as warm-start learning) Zhu et al. (2022); Hou et al. (2019b); Petit et al. (2023); Goswami et al. (2024); Ma et al. (2023). Recently, prompt-based methods have been used in CIL McDonnell et al. (2024); Zhou et al. (2024a), employing linear discriminant analysis Panos et al. (2023) or a simple nearest class mean (NCM) classifier Janson et al. (2022). These methods use a transformer pre-trained model (PTM) on large-scale datasets like ImageNet-21k

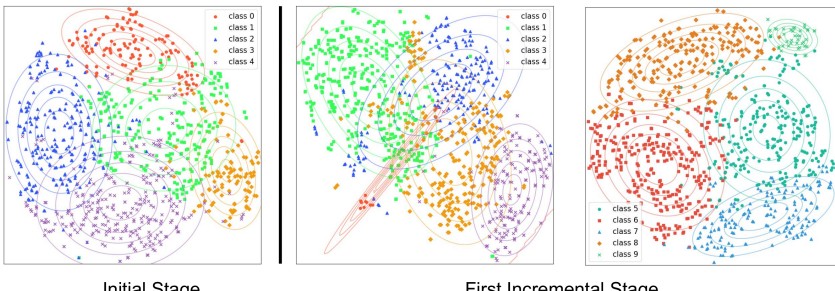

Figure 1: Low-dimensional demonstration of feature space drift (using t-SNE) in the embedding of Task 1's dataset, $\mathcal{D}_1$, while training on Task 2's dataset, $\mathcal{D}_2$. Both tasks contain 5 classes, and the tasks are observed when the simulation is done on CIFAR100/20 up to the first incremental stage.

Ridnik et al. (2021) and focus on classifier-incremental learning. However, the model possesses extreme rigidity and requires high computational cost due to using a task-specific expert.

A category of EFCIL focuses on learning task-specific subspace/prompts and prototypes using either a linear incremental classifier as a classification head or a prototypical network and shows promising results in mitigating forgetting Snell et al. (2017); Cha et al. (2021); Wang et al. (2022b). Expert Gate learns task-wise subspace growing with experts to alleviate forgetting, which is limited by computational cost Aljundi et al. (2017). Wang *et al.* reduced the memory constraint by training a fixed number of experts to generate an ensemble of feature representations but with a high penalty, limiting the plasticity Wang et al. (2022a). Rypeść. *et al.* uses a fixed number of subspaces to learn the Gaussian prototype representation for all the classes and uses an optimal expert to be fine-tuned for the remaining tasks based on statistical measures Rypeść et al. (2023). The major drawbacks with the current prototypes-based approaches are 1) the growing computational cost with task-specific experts and 2) overwriting the respective subspace of the fine-tuned expert by new class prototypes once the new tasks arrive. Figure 1 shows how the task subspace changes incrementally when the model is trained with current methods in the literature. Since the prototype represents the statistical representation of features, a drift in the features results in a severe drift in the prototype, making the model vulnerable to forgetting the previous knowledge.

In this paper, we address these drawbacks by developing a deep mixture of experts (MoE) and leveraging their subspaces to enhance the plasticity of the learning model without task interference during inference. Being inspired by Zhou et al. (2024b), the proposed approach realigns the local semantic relationship between class features in old subspaces into the latest subspace to improve the model's generalization ability. A single expert network from MoE is selected to be fine-tuned on the current task while simultaneously inheriting knowledge from the rest of the experts without these experts explicitly participating in the optimization. We employ a prototype-based ensemble of deep Gaussian classifiers during inference, helping the model estimate the non-linear decision boundary and significantly increasing the model's plasticity. The main contributions are summarized as:

- Introduce a novel *Prototype Consolidation and Ensemble-based Exemplar-Free Deep Incremental Learning (ProCEED)* that leverages the statistics of deep features subspace to realign the representation of old tasks into the latest subspace. ProCEED prevents feature drifts and significantly abridges the stability gap without extra optimization overhead.

- Propose a *knowledge distillation technique* where a single expert (learning model) inherits knowledge from previous tasks learned by MoE without optimizing the entire ensemble of experts on the current task, leading to efficient improvement of the model's plasticity.

- Leverage the subspace ensemble of MoE during inference, which demonstrates superior *task-agnostic* accuracy on challenging benchmark datasets with an equal class distribution across tasks.

## 2    RELATED WORKS

**Class Incremental Learning (CIL)**. CIL approaches can be broadly divided into three categories: rehearsal, regularization, and expansion-based methods. Rehearsal-based methods use representative

samples from previous tasks during training on downstream tasks and apply knowledge distillation to preserve previous knowledge while learning new tasks Hou et al. (2019a); Wu et al. (2019); Wang et al. (2022d); Douillard et al. (2020); Kang et al. (2022). Some of the rehearsal methods mitigated forgetting by controlling the feature adjustment Kirkpatrick et al. (2017); Smith et al. (2021); Toldo & Ozay (2022). Despite the improvements in solving the forgetting issue, rehearsal methods are limited due to the growing computational cost and raise concerns about data privacy.

**Exemplar-Free Class Incremental Learning**. Despite being challenging, learning the sequential tasks without storing samples from previous stages makes a learning agent more pragmatic Li & Hoiem (2016); Yu et al. (2020); Smith et al. (2021); Zhu et al. (2021b); Zhou et al. (2021); Petit et al. (2023). Zhu et al. (2021c); Smith et al. (2021) combined the regularization with prototype rehearsal to enhance the model's plasticity. Prototypes represent the feature space statistics (features mean and standard deviation of respective classes) used to reminiscence the decision boundaries of the previous stages without the explicit need for exemplars. Zhou et al. (2021; 2022b) use prototype augmentation and self-supervision optimization to learn the transferable features for future tasks. Ye & Bors (2020); Cong et al. (2020) trained a generator to rehearse previous knowledge as exemplars. However, due to the need for high-quality generated data, this approach also suffered from forgetting. Learning task-specific prompts with a large pre-trained network as a feature extractor has also received a lot of attention in the literature Wang et al. (2022c;b); McDonnell et al. (2024); Zhou et al. (2024a)

**Dynamic and Ensemble-based Learning** involves neural modifications, including expanding, trimming, or freezing components to suit different incremental tasks Rusu et al. (2016); Yoon et al. (2018); Hung et al. (2019); Li et al. (2019); Nie et al. (2023); Ramesh & Chaudhari (2022). For example, Aljundi et al. (2017) used a dedicated network for each task, while van de Ven et al. (2020) trained separate generative networks for incremental learning stages. The latest CL methods in the literature used a pre-trained ViT as a features extractor and a prototypical network Snell et al. (2017) as a classifier head either by adopting cosine similarity Zhou et al. (2024b) or using random projections McDonnell et al. (2024). Regardless of the massive improvement in accuracy, these approaches require high computational costs and task identity during the inference, which makes these algorithms hardly practical *Class-Incremental Learning*.

**Gaussian Models in CL**. Rehearsal-free CIL methods are vulnerable to recency bias towards the classes of recent tasks due to the cross-entropy loss during optimization Wu et al. (2019); Masana et al. (2022a). Rebuffi et al. (2017); Yu et al. (2020) mitigated this issue by employing the nearest class mean (NCM) classifier with stored class centroids. Rao et al. (2019) modeled the incoming classes with the Gaussian mixture model (GMM). Goswami et al. (2024) adopted the prototypical-based Bayes classifier and inferred the classes using the Mahalanobis distance. These methods require 50% of the samples upfront and require task identity during the inference, which seems unreliable in many practical applications, e.g., medical imaging. To address such issues, Rypeść et al. (2023) incorporated a mixture of experts with multivariate Gaussian distributions to learn the Gaussian prototype of input samples. However, due to the rigid feature-distillation-based regularization, the model focused on maintaining stability over plasticity and required a large number of experts. Furthermore, there is heavy semantic drift and recency bias in the subspace of fine-tuned experts, eventually leading to the problem of forgetting.

# 3 PROCEED: PROTOTYPE CONSOLIDATION AND DEEP ENSEMBLES OF EXPERTS

## 3.1 NOTATIONS AND PROBLEM FORMULATION

In EFCIL, the sequences of $\mathcal{T}$ incremental disjoint tasks arrive from the data distribution $\mathcal{D} = \{\mathcal{D}_t\}_{t=1}^{\mathcal{T}}$. In each stages the data $\mathcal{D}_t = \{\mathcal{X}_t, \mathcal{Y}_t\}$ contains a set of input samples $\mathcal{X}_t = \{x_t^i\}_{i=1}^{N_t}$ and the respective labels $\mathcal{Y}_t = \{y_t^i \in \mathcal{C}_t\}_{j=1}^{N_t}$, where $N_t$ is the number of samples at task $t$, $x_t^i$ represents the $i^{th}$ sample and $\mathcal{C}_t$ is the $t^{th}$ set of labels where $\mathcal{C}_t \cap \mathcal{C}_l = \emptyset$ ($t \neq l$). The model $\mathcal{F} \circ \mathcal{G}_k$ consists of a feature extractor backbone $\mathcal{F}$ with parameters $\boldsymbol{\theta}$ and an incremental Gaussian classifier head $\mathcal{G}_k$ with parameters $\Theta_k$ where, $\Theta_k = (\boldsymbol{\mu}_k, \boldsymbol{\Sigma}_k)$ associated with the $k^{th}$ expert ($k = 1, \cdots, \mathcal{K}$), such that $\mathcal{K} < \mathcal{T}$. At the incremental stage $t$, we append the linear classifier $\mathcal{A}$ with parameters $\phi$ to $\mathcal{F}$ for the optimization. The overall optimization objective is to minimize the empirical loss $\mathcal{L}(.,.)$ with parameters $\boldsymbol{\theta}$ and $\phi$.

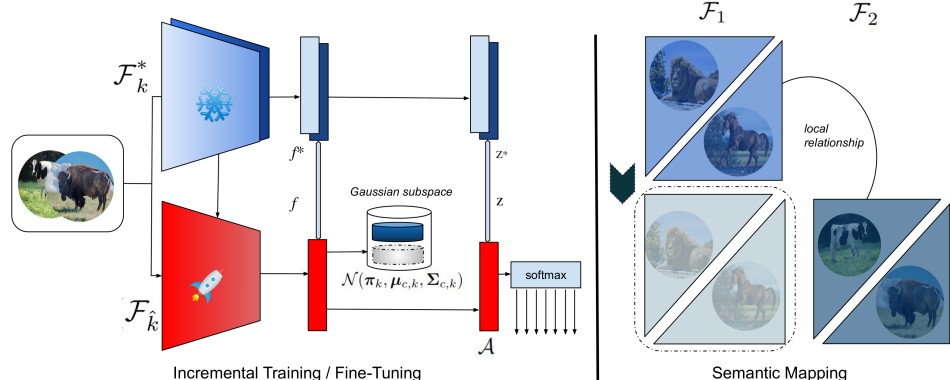

Figure 2: ProCEED comprises $\mathcal{K}$ deep network experts $\mathcal{F} \circ \mathcal{G}_k$ (here $\mathcal{K} = 2$), sharing the initial layers $\mathcal{F}$ for higher computational performance. $\mathcal{F}$ are frozen after the first task. Each expert contains one Gaussian distribution per class $c \in C$ in his unique latent space. In this case, we assume 2 classes for each task. When the second expert, $\mathcal{F}_2$, is trained on task 2, there is severe drift in the subspace of task 1 in the subspace of task 2. After each incremental training, with semantic guided prototype consolidation (right), the subspace of task 1 is realigned in the subspace of task 2.

Mathematically,

$$\arg \min_{\boldsymbol{\theta}, \boldsymbol{\phi}} \sum_{t}^{N_t} \mathbb{E}_{(x_t^i, y_t^i) \sim D_t} \left[ \mathcal{L} \left( y_t^i, \mathcal{A}(\mathcal{F}(x_t^i; \boldsymbol{\theta}); \boldsymbol{\phi}) \right) \right] \tag{1}$$

We formulate an adaptive knowledge distillation-based regularization loss $\mathcal{L}_{KD}$ that allows the knowledge inheritance from all experts while learning a new task. The distillation loss includes a combination of feature and logit distillation. Mathematically, $\mathcal{L}_{KD}$ loss is defined as follows.

$$\mathcal{L}_{KD} = \underbrace{\sum_{\substack{k=1 \\ k \neq \hat{k}}}^{K} \left\| \mathcal{F}_{\hat{k},t}(\mathcal{X}_t; \boldsymbol{\theta}) - \mathcal{F}_{k,t}(\mathcal{X}_t; \boldsymbol{\theta}) \right\|_2^2}_{\text{Feature Distillation}} + \underbrace{\frac{\lambda}{\alpha} \sum_{\substack{k=1 \\ k \neq \hat{k}}}^{K} \mathcal{L}_{CE} \left( \mathcal{A}(\mathcal{F}_{\hat{k},t}(\mathcal{X}_t; \boldsymbol{\theta}), \boldsymbol{\phi}), \mathcal{A}(\mathcal{F}_{k,t}^*(\mathcal{X}_t; \boldsymbol{\theta}), \boldsymbol{\phi}) \right)}_{\text{Logit Distillation}}$$

$$\tag{2}$$

Finally, the total loss function for every task $t$ is formulated as a convex combination of the cross-entropy loss, $\mathcal{L}_{CE}$, on the current task and the adaptive knowledge distillation-based regularization loss $\mathcal{L}_{KD}$. During optimization, while only one best expert is learning the current task, this expert inherits knowledge from the other frozen experts.

$$\mathcal{L}(\boldsymbol{\theta}, \boldsymbol{\phi}; \mathcal{D}_t) = (1 - \alpha)\mathcal{L}_{CE} + \alpha\mathcal{L}_{KD}, \tag{3}$$

The hyper-parameter $\alpha$ controls the trade-off between plasticity and adaptability.

### 3.2 DEEP GAUSSIAN SUBSPACE EXPANSION WITH EXPERTS

We train task-specific experts for tasks $t \leq \mathcal{K}$. To fine-tune the downstream tasks $t > \mathcal{K}$, we adopt the expert that ensures less interference in the embedding space among all experts. After the completion of each incremental training, we remove the linear head and generate the representations of input samples at task $t$ by forwarding them through each expert. The embedding of the $k^{th}$ expert can be represented as

$$f_k = \sum_{t=1}^{\mathcal{T}} \mathcal{F}_k(\mathcal{X}_t; \boldsymbol{\theta}) \tag{4}$$

The Gaussian mixture distribution for $\mathcal{K}$ experts can be written as a linear superposition of the individual Gaussian distribution of each expert.

$$p(f) = \sum_{k=1}^{\mathcal{K}} \pi_k \mathcal{N}(f_k | \Theta_k^c) \tag{5}$$

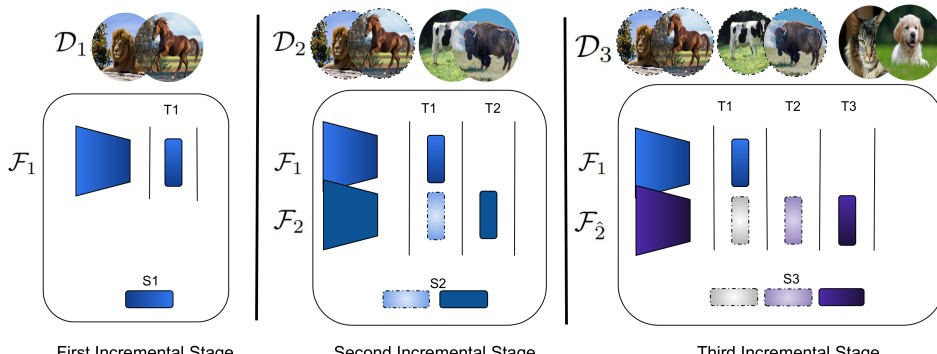

Figure 3: Demonstration of training and realignment steps of ProCEED for each incremental stage. Suppose we have two experts and three incremental tasks with data distribution $\mathcal{D} = \{\mathcal{D}_1, \mathcal{D}_2, \mathcal{D}_3\}$, with two classes in each. The dotted data patterns represent the classes that we only have access to their prototypes in each training stage. We train $T1$ and $T2$ using $\mathcal{F}_1$ and $\mathcal{F}_2$, respectively. After training on the second task, we realign the subspace of $T1$ (denoted by a dashed boundary) with the subspace of $T2$. Suppose we fine-tune $\mathcal{F}_{\hat{2}}$ on the task $T3$; similarly, we align the subspace of $T1$ and $T2$ with the subspace of $T3$.

Equation 5 builds the incremental Gaussian prototypes (respective mean and standard deviation) and is solved using the expectation-maximization (EM) algorithm Bishop (2006a). When a new task arrives, the model learns a new subspace using the respective fine-tuned expert backbone with increasing embedding. The Gaussian prototypes are then memorized in each incremental step that later, during inference, can represent the task-specific information in the Gaussian classifier after each incremental step. Suppose we have two experts $\mathcal{F}_1$ and $\mathcal{F}_2$ available to learn the set of incremental tasks. We train the first expert $\mathcal{F}_1$ on the dataset $\mathcal{D}_1$ in the first stage and approximate the mean prototypes for the classes in $\mathcal{D}_1$, denoted as $\boldsymbol{\mu}_{1,1} = \text{Concat}[\mu_{1,1}^{(1)}, \cdots, \mu_{1,1}^{(|\mathcal{Y}_t|)}]$. The former subscript in $\boldsymbol{\mu}_{1,1}$ is associated with the expert index (first expert) and the latter for the task-specific subspace index (first-task subspace). In the next incremental task, we train the expert $\mathcal{F}_2$ on $\mathcal{D}_2$ and extract the prototypes $\boldsymbol{\mu}_{2,2}$. Since we only have access to $\mathcal{D}_2$ at this stage, we can only compute the prototypes of $\mathcal{D}_2$ in the subspace of $\mathcal{F}_2$. Thus, we can only use the subspace associated with the expert $\mathcal{F}_2$. During inference, we have access to the *drifted prototypes* $\boldsymbol{\mu}_{2,1}$ of the old task along with $\boldsymbol{\mu}_{2,2}$ of the current task in the new embedding subspace of the expert $\mathcal{F}_2$. The prototype $\boldsymbol{\mu}_{2,1}$ does not represent the true distribution of the old task because originally $\mathcal{D}_1$ is trained by the expert $\mathcal{F}_1$. In other words, we need a mechanism to inject the representation of true past prototypes $\boldsymbol{\mu}_{1,1}$ in the latest subspace; otherwise, due to the recency bias, the Gaussian classifier is vulnerable to *Catastrophic Forgetting*. To learn the classes of the $\mathcal{D}_3$, we need to select the expert (either $\mathcal{F}_1$ or $\mathcal{F}_2$) that coincides less with the current task's features and follow a similar approach by Rypeść et al. (2023). Suppose the expert $\mathcal{F}_2$ is selected to be fine-tuned on $\mathcal{D}_3$, similarly, we need to realign the prototypes of $\mathcal{D}_1$ and $\mathcal{D}_2$ in the new subspace of $\mathcal{F}_2$ learned from $\mathcal{D}_3$. For $\mathcal{K}$ experts with $\mathcal{T}$ incremental tasks, the prototypes of the Gaussian classifier during inference can be arranged in the following matrix $\mathcal{G}$.

$$
\mathcal{G} = \begin{bmatrix} \boldsymbol{\mu}_{1,1} & 0 & \cdots & 0 \\ \boldsymbol{\mu}_{2,1} & \boldsymbol{\mu}_{2,2} & \cdots & 0 \\ \vdots & \vdots & \ddots & \vdots \\ \boldsymbol{\mu}_{\mathcal{K},1} & \boldsymbol{\mu}_{\mathcal{K},2} & \cdots & \boldsymbol{\mu}_{\mathcal{K},\mathcal{T}} \end{bmatrix} \tag{6}
$$

### 3.3 ANGULAR DRIFT COMPENSATION VIA SEMANTIC MAPPING

The matrix $\mathcal{G}$ in Equation 6 represents the task-subspace specific prototypes $\boldsymbol{\mu}$ of the Gaussian classifier. However, when we fine-tune an expert $i$ on a new task $t$, we need to realign the previous prototypes $(\boldsymbol{\mu}_{i,1}, \boldsymbol{\mu}_{i,2}, \cdots, \boldsymbol{\mu}_{i,t-1})$ learned from all previous tasks in the new subspace of the fine-tuned expert. It is important to highlight that the experts do not have access to any samples from prior tasks. In other words, the entries below the diagonal of the matrix $\mathcal{G}$ in Equation 6 need to be realigned in the new subspace of prototypes of the latest task.

Without loss of generality, we formulate the above misalignment issue such that given all prototypes (old $o$ and new $n$) associated with any expert, the target is to project the old class prototypes onto the latest subspace to obtain a new realigned prototype $\hat{\boldsymbol{\mu}}_{n,o}$ using $\boldsymbol{\mu}_{o,o}$, $\boldsymbol{\mu}_{n,o}$, and $\boldsymbol{\mu}_{n,n}$. Intuitively, prototypes of similar classes contain similar feature representations to infer the labels of those classes. For instance, representative features for a 'dog' also contain features to represent the 'fox'. We take into account that this semantic similarity can be shared among different sub-spaces of various classes. Therefore, we propose to compute *semantic information* in the co-occurrence space and realign the prototypes by projecting them into respective sub-spaces. Specifically, we measure the cosine similarity between prototypes of previous tasks in both old and new sub-spaces, i.e., $\boldsymbol{\mu}_{o,o}$ and $\boldsymbol{\mu}_{n,o}$, respectively, and utilize it to project the prototypes in the new embedding space. The classes with similarity among all classes are calculated using prototypes in the co-occurrences subspace:

$$\text{Sim}_{i,j} = \frac{\boldsymbol{\mu}_{o,o}[i]}{\|\boldsymbol{\mu}_{o,o}[i]\|_2} \cdot \frac{\boldsymbol{\mu}_{n,o}[j]^\top}{\|\boldsymbol{\mu}_{n,o}[j]\|_2}, \tag{7}$$

where the index $i$ represents the $i^{th}$ class prototype. Equation 7 further undergoes the softmax normalization: $\text{Sim}_{i,j} = \frac{\exp(\text{Sim}_{i,j})}{\sum_j \exp(\text{Sim}_{i,j})}$. The normalized similarity holds the local relationship between the old classes subspace and the new subspace in co-occurrence spaces.

Once we obtain the local relationship between the subspace of experts using the normalized cosine similarity, we inject this similarity information into the prototypes of old classes to realign them in the new subspace. The transformed prototype of old classes into new subspaces can be measured as a weighted combination of new and old class prototypes:

$$\hat{\boldsymbol{\mu}}_{n,o}[i] = \boldsymbol{\mu}_{n,o}[i] + \sum_j \text{Sim}_{i,j} \times \boldsymbol{\mu}_{n,n}[j]. \tag{8}$$

After semantic mapping, the updated prototype matrix $\mathcal{G}$ of the Gaussian classifier is updated as the following:

$$\mathcal{G} = \begin{bmatrix} \boldsymbol{\mu}_{1,1} & 0 & \cdots & 0 \\ \hat{\boldsymbol{\mu}}_{2,1} & \boldsymbol{\mu}_{2,2} & \cdots & 0 \\ \vdots & \hat{\boldsymbol{\mu}}_{3,2} & \ddots & \vdots \\ \hat{\boldsymbol{\mu}}_{\mathcal{K},1} & \hat{\boldsymbol{\mu}}_{\mathcal{K},2} & \cdots & \boldsymbol{\mu}_{\mathcal{K},\mathcal{T}} \end{bmatrix} \tag{9}$$

### 3.4 INFERENCE VIA DEEP SUBSPACE ENSEMBLE

At this point, we have introduced how the Gaussian subspace expands and gets updates in incremental stages after each training or fine-tuning session. During inference, we compute the latent space features of input samples $f = \mathcal{F}_{\hat{k}}(x_{t,j}; \theta)$. The logit of task sample $x_t$, which is fine-tuned in expert $k$, can be expressed using the log-likelihood expectation of Gaussian mixture distribution.

$$\ln p(f|\boldsymbol{\pi}, \boldsymbol{\mu}_k, \boldsymbol{\Sigma}_k) = \sum^{N_t} \ln \{\pi_k \mathcal{N}(\mathcal{F}(\mathcal{X}_t; \theta)|\Theta_k)\}, \tag{10}$$

where $\boldsymbol{\pi}$ is the vector of the mixing coefficients, $\Theta_k = (\boldsymbol{\mu}_k, \boldsymbol{\Sigma}_k)$. Here, $\boldsymbol{\Sigma}_k$ is the covariance matrix, and $\boldsymbol{\mu}_k$ is the mean vector, initialized using the K-Means algorithm. For $k^{th}$ expert, the maximum likelihood estimator (MLE) solution of Equation 10 is derived using the expectation maximization (EM) algorithm Bishop (2006b) and can be expressed as the following:

$$l_k(f|\boldsymbol{\mu}_k, \boldsymbol{\Sigma}_k) = -\frac{1}{2}\left[\ln(|\boldsymbol{\Sigma}_k|) + N\ln(2\pi) + (f_k - \boldsymbol{\mu}_k)^T(\boldsymbol{\Sigma}_k)^{-1}(f_k - \boldsymbol{\mu}_k)\right] \tag{11}$$

where $N$ is the dimension of latent space feature representation. The softmax values of the maximum-likelihood probabilities (logits) for each expert, i.e., $\hat{l}_k^1, \cdots, \hat{l}_k^{|C|}$, are then computed with the temperature parameter $\tau$, where $C$ is the set of classes seen so far, i.e., softmax$(l_k^1, \cdots, l_k^{|C|}; \tau)$. For *task-agnostic* inference, we compute an average of all experts, and the predicted class $c$ is the one with the highest expected value $\mathbb{E}[l_k^c]$. Figure 4 also gives a visual explanation for the ensemble inference process of ProCEED.

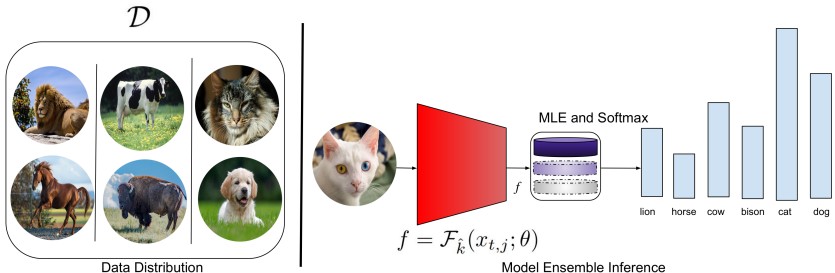

Figure 4: Once training on data $\mathcal{D}$ is complete, during inference, we calculate the latent representation $f = \mathcal{F}_{\hat{k}}(x_{t,j};\theta)$ of the test samples from respective tasks. We then compute the log-likelihood of the features using Equation 10 in each subspace. After softmax-normalization of likelihood probabilities, we compute the mean of each subspace, and the argument with the highest mean is the *task-agnostic* inferred class.

## 4 EXPERIMENTS

In this section, we experiment with four benchmark datasets and compare the proposed model with the state-of-the-art (SOTA) methods to validate the incremental learning ability.

### 4.1 EXPERIMENTAL SETUP

**CIL Datasets:** We conduct the experiments using CIFAR100 (100 classes) Krizhevsky (2009), ImageNetSubset100 (100 classes) Deng et al. (2009), TinyImageNet200 (200 classes) Le & Yang (2015), and DomainNet (345 classes from 6 domains) Peng et al. (2019). These datasets possess typical CIL benchmarks and out-of-distribution with severe drift in inter-domain distribution allowing us to assess the robustness of the model against domain drift.

**Dataset Split:** Each dataset is split into an equal number of classes in each task from the beginning. This approach is more challenging due to a weaker backbone due to fewer classes in the initial tasks. We reproduce the results using FACIL Masana et al. (2022b) and PyCIL Zhou et al. (2021) frameworks.

**Compared Baselines:** We compare the proposed framework against several CIL approaches Kirkpatrick et al. (2017); Li & Hoiem (2016); Zhu et al. (2021b); Hou et al. (2019a); Zhou et al. (2022a); Zhu et al. (2021a); Petit et al. (2023); Rypeść et al. (2023); Magistri et al. (2024); Goswami et al. (2024). We run the experiments in three exemplar-free learning scenarios: *cold-start* (classes are split evenly in all incremental steps), *warm-start* (initial task contains 50% of the total classes, and the rest of the classes are split evenly) and the *task-aware incremental setting*, where task-id is available during the inference. All baselines are reproduced either from the official implementation or FACIL Masana et al. (2022b) and PyCIL Zhou et al. (2021) frameworks.

**Implementation Details:** We evaluate our algorithm based on the FACIL framework Masana et al. (2022b) for both class and domain incremental learning (CIL and DIL). For all simulations, we train a ResNet-32 architecture He et al. (2016b) from scratch as a feature-extractor network with stochastic gradient descent (SGD) and an initial learning rate of 0.05. The hyperparameters $\alpha$ and $\lambda$ in Equations 2 and 3 are set to 0.99 and 1, respectively.

**Evaluation Metric:** We use the average incremental accuracy, $A_t$, defined as the average accuracy across the first $t$ tasks after incremental training on these tasks.

$$A_t \triangleq \frac{1}{\mathcal{T}} \sum_{i=1}^{\mathcal{T}} A_i. \tag{12}$$

More details on the evaluation are provided in the Appendix A.5 section.

### 4.2 SIMULATION RESULTS

**Cold-Start Learning:** In Table 1, we present a detailed comparison of ProCEED and the state-of-the-art exemplar-free CL models for CIFAR100, TinyImageNet200, ImageNetSubset100, and

Table 1: Average incremental task-agnostic accuracy (%) for exemplar-free CL with different number of incremental tasks evaluated on CIFAR100, TinyImageNet200, ImageNetSubset100, and DomainNet using a *cold-start* scenario. The best results are in **bold**, and the second best is underlined

| Approach | CIFAR100 | | | TinyImageNet | | ImageNetSubset | | DomainNet | | |
|---|---|---|---|---|---|---|---|---|---|---|
| | $T$=10 | $T$=20 | $T$=50 | $T$=10 | $T$=20 | $T$=10 | $T$=20 | $T$=12 | $T$=24 | $T$=36 |
| Finetuning | 24.44 | 17.84 | 7.18 | 21.28 | 14.35 | 26.48 | 18.15 | 19.78 | 15.33 | 11.93 |
| EWC Kirkpatrick et al. (2017) | 31.31 | 22.74 | 10.33 | 21.14 | 14.55 | 27.77 | 18.52 | 18.94 | 13.83 | 11.82 |
| LwF Li & Hoiem (2016) | 39.24 | 29.24 | 14.24 | 23.61 | 17.21 | 45.02 | 34.63 | 19.54 | 11.66 | 11.66 |
| LUCIR Hou et al. (2019a) | 36.47 | 22.98 | 10.71 | 25.21 | 17.73 | 35.07 | 21.69 | 20.07 | 13.56 | 10.52 |
| IL2A Zhu et al. (2021a) | 37.96 | 40.63 | 39.98 | 43.75 | 30.89 | – | – | 18.54 | 16.74 | 15.34 |
| PASS Zhu et al. (2021b) | 36.48 | 41.99 | 40.54 | 47.11 | 34.92 | 50.56 | 43.04 | 25.56 | 21.45 | 11.26 |
| SSRE Zhu et al. (2022) | 42.25 | 30.59 | 30.18 | 46.34 | 43.56 | 42.98 | 31.66 | 25.79 | 20.31 | 20.45 |
| FeTrIL Petit et al. (2023) | 41.55 | 38.34 | 34.73 | 51.57 | 45.09 | 44.56 | 35.37 | 37.32 | 31.76 | 30.14 |
| SEED Rypeść et al. (2023) | 60.71 | 55.25 | 32.72 | 46.92 | 39.39 | 65.72 | 63.71 | 44.64 | 34.32 | 30.12 |
| EFC Magistri et al. (2024) | 60.56 | 52.65 | 29.36 | 38.85 | 33.15 | 60.85 | 55.34 | – | – | – |
| FeCAM Goswami et al. (2024) | 61.72 | 58.75 | 37.55 | 46.34 | 40.85 | 58.03 | 44.73 | – | – | – |
| ProCEED$^{Bayes}$A.8 | 71.00 | 58.79 | 49.62 | 46.69 | 47.09 | 65.32 | 68.55 | 46.53 | 47.59 | 43.17 |
| ProCEED$^{MLE}$ | **74.56** | **63.23** | **53.34** | **52.00** | **51.15** | **72.55** | **71.46** | **52.33** | **51.15** | **51.34** |
| Joint (Oracle) | 79.00 | 79.52 | 80.77 | 67.74 | 69.34 | 83.23 | 84.64 | 64.08 | 65.43 | 69.72 |

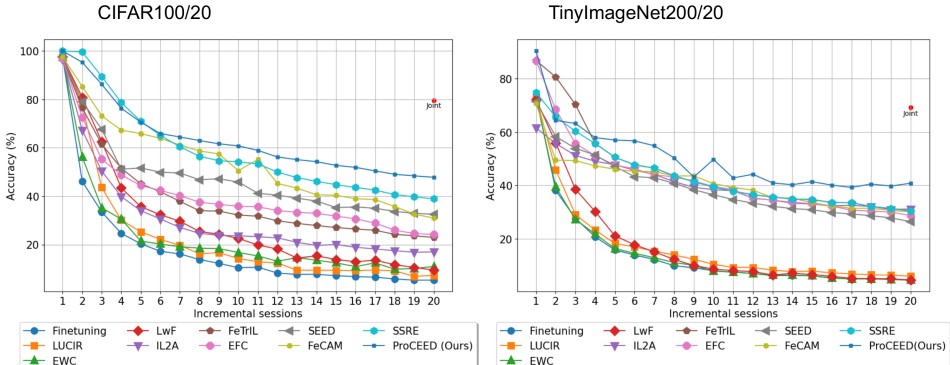

Figure 5: Average incremental accuracy measured after each task in two scenarios: (1) CIFAR100 (left) splits into 20 tasks with 5 classes each, and (2) TinyImageNet (right) splits into 20 tasks with 10 classes each.

DomainNet datasets. We report the average incremental accuracy for different splitting conditions and domain shifts. In each splitting condition for various datasets, ProCEED produces superior performance compared to other methods by a significant margin. For CIFAR100 ($T = 10$ and $T = 50$), ProCEED outperforms the second-best method FeCAM Goswami et al. (2024) and PASS Zhu et al. (2021b), respectively, by 13%. Table 1 shows that the results are coherent when using CIFAR100, ImageNetSubset100, or TinyImageNet200 datasets for all rehearsal-free methods. ProCEED consistently achieves the best accuracy (or the second best for $T = 10$ in TinyImageNet) as compared to all other methods in the literature. From the result of DomainNet, we can conclude that ProCEED is robust to distributional shift and possesses more plasticity compared to other methods. An important observation for DomainNet, for $T = 24$ and $T = 36$, is that ProCEED performs exceptionally higher compared to the second-best methods Rypeść et al. (2023); Petit et al. (2023) by 17.46% and 20.98% points, respectively. Moreover, ProCEED maintains its accuracy regardless of the increase in the number of tasks. In contrast, the performance of all other approaches decreases when the number of tasks increases. We also present the joint optimization as an upper bound for CIL. Furthermore, for equal splits (cold-start scenarios), the detailed accuracy for CIFAR100 and TinyImageNet200 is presented in Figure 5. This figure shows that ProCEED demonstrates higher knowledge retention from previous tasks even if very little data is provided in the initial task compared to approaches that only employ parameter or feature-regularized-based distillation. After 20 incremental learning sessions using the CIFAR100 dataset (left graph), ProCEED performs 8.45% points higher than the second-best FeCAM method Goswami et al. (2024) and 10.42% points higher than the second-best approach SSRE Zhu et al. (2022) for TinyImageNet200 (right graph).

**Warm-Start Learning:** We evaluate ProCEED by initializing the backbone feature extractor with 50% of the total classes in the first task and evenly distributing the remaining classes across subsequent tasks. Detailed results are presented in Table 2. For CIFAR100, ProCEED outperforms the second-best methods, SEED Rypeść et al. (2023) and FeTrIL Petit et al. (2023), by 5.43%, 1.33%, and

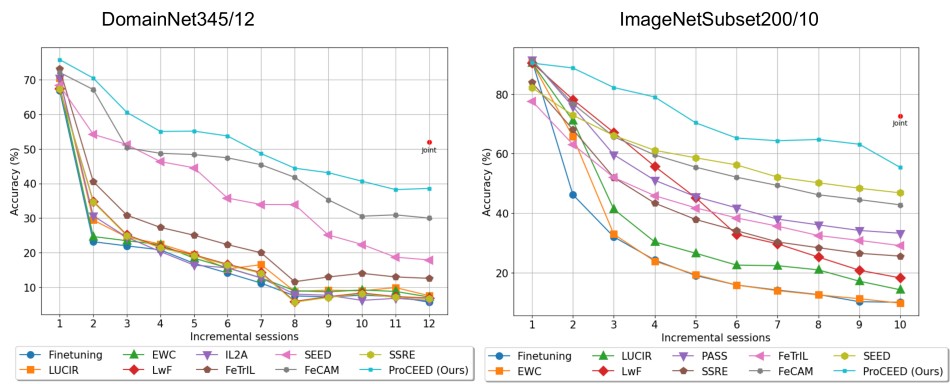

Figure 6: Average incremental accuracy measured after each task in two scenarios: (1) DomainNet (left) splits into 12 tasks, and (2) ImageNet-Subset (right) splits into 10 tasks.

Table 2: Average incremental *task-agnostic* accuracy (%) for rehearsal-free CL with different number of incremental tasks evaluated on CIFAR100, TinyImageNet-200, and ImageNetSubset-100 using a *warm-start* scenario. The best results are **in bold**, and the second best is underlined.

| Appraoch | CIFAR100 | | | TinyImageNet | | | ImageNetSubset | | |
|---|---|---|---|---|---|---|---|---|---|
| | $T$=6 | $T$=11 | $T$=21 | $T$=6 | $T$=11 | $T$=21 | $T$=6 | $T$=11 | $T$=21 |
| EWC Kirkpatrick et al. (2017) | 22.56 | 21.34 | 18.67 | 15.83 | 12.54 | 10.32 | 27.16 | 22.39 | 20.43 |
| LwF Li & Hoiem (2016) | 43.94 | 27.45 | 20.23 | 23.21 | 17.55 | 15.33 | 44.62 | 40.45 | 40.01 |
| DeeSIL Belouadah & Popescu (2018) | 55.43 | 45.32 | 35.86 | 41.55 | 32.34 | 29.43 | 65.43 | 58.49 | 45.45 |
| PASS* Zhu et al. (2021b) | 63.84 | 61.81 | 57.43 | 40.32 | 35.65 | 25.65 | 64.44 | 61.86 | 51.35 |
| IL2A Zhu et al. (2021a) | 65.21 | 58.39 | 50.56 | 45.23 | 42.22 | 37.45 | 62.42 | 60.34 | 55.65 |
| SSRE Zhu et al. (2022) | 64.32 | 64.21 | 60.64 | 49.52 | 45.62 | 45.54 | 68.76 | 65.85 | 60.43 |
| FeTrIL Petit et al. (2023) | 66.45 | 65.61 | 61.77 | 54.34 | 52.67 | 52.45 | 68.54 | 67.63 | 66.54 |
| EFC Magistri et al. (2024) | 68.85* | 62.17 | 58.54 | 50.41 | 48.87 | 48.32 | 50.46 | 48.63 | 48.64 |
| FeCAM Goswami et al. (2024) | 68.45 | 68.94 | 60.65 | 65.39 | 60.23 | 55.58 | 58.56 | 60.41 | 59.34 |
| SEED Rypeść et al. (2023) | 72.13 | 69.35 | 58.03 | 62.68 | 61.37 | **61.45** | **70.54** | 65.55 | 63.42 |
| ProCEED$^{MLE}$ (ours) | **77.54** | **70.67** | **66.42** | **71.44** | **64.93** | 61.64 | 69.23 | **71.11** | **66.12** |

5.22% points for $T = 6$, 11, and 21, respectively. With a frozen feature extractor, ProCEED performs comparably to SEED's best result on TinyImageNet200, with a margin of 1.19% for $T = 21$, and shows similar trends on ImageNetSubset200 for $T = 6$. The results indicate that the weight-regularization methods, such as EWC and LwF, exhibit poor accuracy, while knowledge-distillation-based methods, including IL2A, SSRE, EFC, SEED, FeCAM, and ProCEED, achieve superior average incremental accuracy.

**Experiments with Pre-trained Weights:** Table 4 shows a comparison between different exemplar-free CL methods by using the pre-trained weights of the ResNet-32 model He et al. (2016a). We compute the model's performance for CIFAR100, TinyImageNet200, and ImageNetSubset100. We also report the accuracy when replaying all the samples from the previous task as Joint (Upper bound) for the pre-trained weights. ProCEED shows better performance than other models for CIFAR100 and TinyImageNet200. Furthermore, in Table 6, we compare the proposed ProCEED with SOTA PTM-based subspace expansion methods Zhou et al. (2024b); McDonnell et al. (2024) that use pre-trained ViT trained on ImageNet21K and

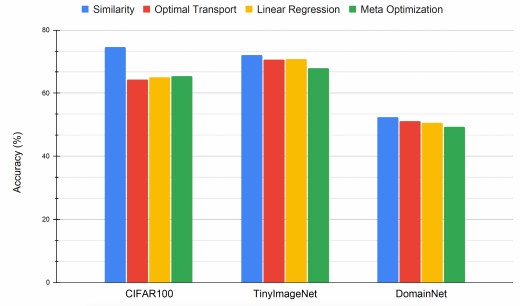

Figure 7: Accuracy of the proposed framework with variations of drift compensation strategies. Similarity with angular drift performs the best among all variations.

we report the accuracy. ProCEED still shows its supremacy for *task-aware* evaluation.

**Ablation Analysis:** Table 3 shows the detailed breakdown of the performance of ProCEED on CIFAR100 using ResNet-32 He et al. (2016a). We evaluate the ProCEED model performance after ablating various components of the objective function as represented in Equations 2 and 3. The symbol (✓) indicates that the operation is applied, while ("x") denotes its absence. Each of the

Table 3: Demonstration of the impact of individual model's components on accuracy (%) within our proposed method, ProCEED, on exemplar-free CIL across 10 tasks of CIFAR100 and TinyImageNet datasets evaluated using the *task-agnostic* settings.

| Cov. Matrix | Logits Distillation [2] | Feature Distillation [2] | Drift Compensation [8] | CIFAR100/10 | | TinyImageNet200/10 | |
|---|---|---|---|---|---|---|---|
| | | | | LTAg Acc | TAg Acc | LTAg Acc | TAg Acc |
| Diag | x | ✓ | x | 52.63 | 63.58 | 30.47 | 44.22 |
| Diag | ✓ | x | x | 53.44 | 66.06 | 35.02 | 45.04 |
| Diag | ✓ | ✓ | x | 54.44 | 66.60 | 39.03 | 46.48 |
| Diag | ✓ | ✓ | ✓ | 58.16 | 64.77 | 40.15 | 46.43 |
| Full | x | ✓ | x | 60.71 | 65.63 | 43.68 | 47.92 |
| Full | ✓ | x | x | 60.37 | 65.38 | 46.55 | 47.37 |
| Full | ✓ | ✓ | x | 61.22 | 67.99 | 47.12 | 48.32 |
| Full | ✓ | ✓ | ✓ | **67.54** | **74.56** | **48.33** | **52.00** |

Table 4: Average incremental *task-agnostic* (TAg) accuracy (%) at the end of the incremental session with pre-trained ResNet-32.

| Approach | CIFAR100 TAg Acc | TinyImageNet TAg Acc | ImageNetSubset TAg Acc |
|---|---|---|---|
| EWC | 45.00 | 35.00 | 35.38 |
| LwF | 45.71 | 30.50 | 50.53 |
| FetrIL | 48.53 | 56.61 | 50.39 |
| FeCAM | 50.34 | 63.77 | 55.56 |
| SEED | 63.32 | 55.47 | **69.44** |
| EFC | 62.73 | 55.73 | 60.43 |
| ProCEED | **68.50** | **65.33** | 64.61 |
| Joint | 88.90 | 75.10 | 75.23 |

Table 5: Average accuracy of ProCEED compared to the other methods for CIFAR100/10 dataset with a varying number of experts

| Approach | 5 | 4 | 3 | 2 | 1 |
|---|---|---|---|---|---|
| CoSCL | 57.33 | 50.12 | 40.59 | 35.31 | 30.78 |
| SEED | 60.61 | 55.32 | 45.62 | 40.32 | 40.33 |
| ProCEED | 74.56 | 73.27 | 70.71 | 68.40 | 67.13 |

Table 6: Average *task-aware* (TAw) and Last Iterate (LTAw) accuracy (%) of our method compared to recent methods using 500 exemplars on ResNet-32 He et al. (2016a). We show the number of shared parameters (in millions) by (Par). The methods with † use a pre-trained ViT as a feature extractor.

| Approach | Par | Ex. | CIFAR100 | | ImageNetSubset | |
|---|---|---|---|---|---|---|
| | | | TAw Acc | LTAw Acc | TAw Acc | LTAw Acc |
| iCaRL* | 9.2 | ✓ | 65.4 | 56.3 | 62.6 | 53.7 |
| DER* | 9.2 | ✓ | 73.2 | 66.2 | 77.6 | 71.1 |
| PODNet* | 6.8 | ✓ | 67.8 | 57.6 | 73.8 | 62.9 |
| Coil* | 6.8 | ✓ | – | – | 59.8 | 43.4 |
| WA* | 6.8 | ✓ | 69.9 | 61.5 | 65.8 | 56.6 |
| BiC | 6.8 | ✓ | 66.14 | 55.36 | 66.43 | 49.92 |
| FOSTER | 6.8 | ✓ | 67.92 | 60.24 | 69.94 | 63.1 |
| MEMO | 5.4 | ✓ | – | – | 76.74 | 70.21 |
| FeCAM | 4.7 | x | 70.99 | 62.13 | 78.43 | 73.09 |
| SEED | 3.2 | x | 86.54 | 90.32 | 75.54 | 69.56 |
| EASE† | 88 | x | 91.56 | 85.35 | 76.34 | 70.39 |
| RanPAC† | 88 | x | 92.20 | **91.01** | 77.01 | 60.34 |
| ProCEED | 3.2 | x | **92.45** | 88.34 | **80.27** | **73.56** |

components plays a vital role in balancing the *stability-plasticity* dilemma. By analyzing the results in Table 3, we notice that applying the subspace realignment preserves the stability of the network by mitigating the angular feature drift. Furthermore, to compare the effectiveness of the proposed drift compensation network, we also simulate other realignment methods, such as optimal transport Courty et al. (2017), Nejjar et al. (2023), meta optimization Finn et al. (2017), to realign the subspace of the previous tasks. Figure 7 shows that the similarity-based angular drift compensation performs the best among other variations.

## 5 CONCLUSIONS

The practicality of real-world learning agents is fulfilled by the ability of their inherent models to learn incrementally. This paper proposes a Prototype Consolidation and subspace Ensemble Exemplar-free Deep class-incremental learning (ProCEED) with a CNN-based mixture of experts (MoE). The proposed ProCEED prevents a drift in the representation of previous tasks by consolidating the previous knowledge with the current task knowledge. The drift compensation is achieved by mapping the local semantic relationship between features of previous tasks and the current one, eventually significantly reducing the forgetting. The model also leverages adaptive knowledge distillation as a regularizer that seeks to inherit the learned features representation from MoE without causing extra computational overhead and much fewer parameters. ProCEED demonstrates superior improvement in accuracy compared to state-of-the-art CIL methods when empirically validated on challenging benchmark datasets in *cold-start*, *task-agnostic*, and *exemplar-free* settings.

**Limitations:** The limitations and future directions of the proposed method are summarized as follows: (1) Estimating the feature covariance drift is still an open question and is a potential future extension, which would better estimate the exact drift in the representations. (2) Approximating the prototype of the old task associated with experts, specifically, completing the element above the diagonal elements in Equation 9, could significantly enhance the model plasticity.

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
