# OpenReview forum: "ProCEED: Prototype Consolidation and Ensemble-based Exemplar-Free Deep Incremental Learning"
_ICLR.cc/2025/Conference — ICLR 2025 Conference Withdrawn Submission_

### Official Review · Reviewer_FscT · 2024-10-20

**Soundness:** 1
**Presentation:** 1
**Contribution:** 1
**Rating:** 1
**Confidence:** 5

**Summary:**

The paper addresses the problem of exemplar-free class incremental learning by proposing a method called  ProCEED, which leverages an ensemble of mixture of experts (MoE).

Given a fixed number of experts $K$, ProCEED fine-tunes only one expert per task. When the number of encountered tasks $T$ is less than $K$, the experts, which are randomly initialized, are sequentially trained. When $T$ exceeds $K$, the expert with the least overlap with the distribution of new classes in its latent space is chosen for fine-tuning. The criterion for measuring the overlap of new classes with respect to the old classes' distribution in each expert is based on the symmetrized Kullback-Leibler divergence. Regarding the training loss of each expert, the  standard cross-entropy loss along with a knowledge distillation loss is employed. The distillation is applied to both the features (feature distillation) and the logits (logit distillation), with the goal that one expert benefits from the knowledge of the other frozen experts.  At inference time, all the experts make predictions using a Gaussian log-likelihood, parameterized by class means and covariances. The final prediction is computed as the average of all experts, with the class $c$  that has the highest expectation being selected.

As the number of tasks increases and $T > K$, fine-tuning each expert modifies the feature representations, which consequently affects the old class representations (for which there is no more data). To update the old class means (or prototypes) for the Gaussian-based classifier, the authors apply a semantic mapping technique, which aims to project the old prototypes into the new embedding space.

The method show state-of-the-art results.

**Strengths:**

- The use of both feature distillation and logit distillation for training loss yields better results compared to employing only one of them, when the semantic mapping is not applied (as shown in Table 3 experiments).
- On CIFAR-100, the method appears robust even with a lower number of experts (as shown in Table 5 experiments).
- The method demonstrates strong performance (as shown in Table 1, 2 experiments).

**Weaknesses:**

The proposed method, ProCEED, appears to be a blend of two existing incremental learning methods, SEED and EASE, published in recent conference papers with minimal changes:

-  **SEED (ICLR 2024):**  [https://openreview.net/pdf?id=sSyytcewxe](https://openreview.net/pdf?id=sSyytcewxe)
- **EASE (CVPR 2024):** [https://openaccess.thecvf.com/content/CVPR2024/papers/Zhou_Expandable_Subspace_Ensemble_for_Pre-Trained_Model-Based_Class-Incremental_Learning_CVPR_2024_paper.pdf](https://openaccess.thecvf.com/content/CVPR2024/papers/Zhou_Expandable_Subspace_Ensemble_for_Pre-Trained_Model-Based_Class-Incremental_Learning_CVPR_2024_paper.pdf)

While the submission does cite these papers (which is acceptable), the authors have not sufficiently differentiated their work from these prior approaches, raising **fairness** concerns regarding the proper crediting of the original contributions from these works. Moreover, there is significant **textual overlap**  between SEED and ProCEED, as well as between EASE and ProCEED, with only minor rephrasing and notation changes. Finally, since no relevant contribution in this paper appears to be original, the paper **lacks novelty**. Below, I go further details on my major concerns:

### Fairness Concerns
The main components of ProCEED are the Deep Ensemble of Experts, Regularization Loss for Expert Training, and Angular Drift Compensation via Semantic Mapping.

As for the first two components, they are exactly the same as those described in the SEED paper, except for the addition of Logit Distillation to feature distillation in **Equation (2)** (in SEED, the regularization loss with only feature distillation appears at the top of page 5). In the current paper, the relationship between SEED and the proposed ProCEED method is unclear. The only acknowledgment is a mention on page 5, line 253,  that SEED’s methodology is used to identify overlapping classes, but this is presented in a very generic way. However, this is not the only aspect of SEED that ProCEED uses: **Section 3.2** (up until **Equation (5)**) and **Section 3.4** directly implement SEED’s methodology without any modification (the citation of SEED should be explicitly included at these points in the text). Additionally, the regularization loss in **Equation (2)** is the same as SEED's, with the addition of Logit Distillation, a distinction that should be clearly identified in **Section 3.1**.

As for Angular Drift Compensation via Semantic Mapping (described in the second half of **Section 3.2**, after Equation (5), and in **Section 3.3**), this is the same strategy used by the EASE approach, with a similar name and performing the same task. However, EASE is never mentioned in those sections. The only reference putting in relationship ProCEED and EASE,  it appears as a simple "Being Inspired by" at the end of the introduction, at line 87, without any explicit mention in the rest of the paper. There is no reference to EASE in **Section 3.2** or **Section 3.3**, compared to the proposed approach.

Finally, Figures 2, 3, and 4 lack acknowledgements in the captions. Even if the formatting of these figures is original to this work, they are merely explaining EASE and SEED. Figure 2 (left) explains SEED, while Figure 2 (right) illustrates the semantic mapping of EASE. Similarly, Figure 3 provides more details about the semantic mapping of EASE, and Figure 4 explains the inference process of SEED.

### Multiple Text Overlaps
While the review process is double-blind, I want to highlight the **significant textual overlap** between this submission and the two prior works of SEED and EASE papers. It’s important to note that, due to the double-blind nature of the review, I cannot definitively confirm the identity of the authors and thus cannot conclusively claim plagiarism. However, based on the analysis of the following sections, there are strong similarities that warrant closer attention.  Large portions of Sections **3.2**, **3.3**, and **3.4** are closely aligned with these prior works, with only minor rephrasing and changes in notation. Below, I provide further details on the textual overlap:

- **Section 3.2**, after Equation (5), is a simple rephrasing of Section 4.2 of the EASE paper up until the phrase "Without loss of generality."

- **Section 3.3** of the submission, which the authors repeatedly claim as a key contribution ("we propose"), is almost identical to Section 4.2 of the EASE paper, starting from the phrase "Without loss of generality." Only minor rewording and notational changes are performed (e.g., swapping "lion" and "cat" for "dog" and "fox").
- **Section 3.4** closely mirrors Section 3 of the SEED paper (Algorithm), with similar rephrasing but retaining the same core content and very similar notations.

### Lack of Novelty

As mentioned above, there is no methodology or textual section in the present paper that offers a genuinely novel contribution. While the addition of logit distillation slightly improves performance compared to using only feature distillation (as in SEED), as noted in the strengths section, it is a minor novelty and not substantial enough for publication. Logit distillation is already a well-established technique in the incremental learning literature (see, for example, the XDER or FACIL papers). Simply incorporating it here does not introduce any significant novelty.

Furthermore, the use of EASE’s methodology in incremental learning—originally employed in the CVPR paper for aligning prototypes in a pre-trained model with multiple adapters, which can be viewed as a multiple experts model—does not introduce novelty when applied within the SEED approach for exemplar-free cold start incremental learning with a fixed number of experts. Transferring a methodology from one incremental context (EASE) to another incremental context (SEED) without substantial innovation or modification does not constitute a meaningful novel contribution.

Even the experimental methodology is not entirely novel. Aside from adding more competitors compared to the SEED paper and showing robustness with fewer experts on a single dataset (as highlighted in the strengths section), the ablation study in **Figure 7**, which discusses different methods for updating the prototype, is the same ablation study as in the EASE paper (Figure 7(b), CVPR paper), just on  different datasets.

**Questions:**

The current submission presents several **fairness**  concerns, arising from the failure to properly cite prior works, EASE (CVPR 2024)  and SEED (ICLR 2024), that are integral to the proposed approach. Moreover the **significant textual overlap** with the EASE  and SEED  papers, outlined in the weaknesses section, raises  concerns of potential plagiarism that warrant closer attention. Finally, the submission lacks novelty, as there is no original methodological contribution beyond the addition of logit distillation to the SEED loss and the transfer of EASE methodology in the SEED context (as outlined in the weaknesses section). While the proposed approach shows good performance, the serious issues related to fairness and significant textual overlap in the methodology raise doubts about the validity of the experimental section as well. Therefore, my recommendation is strong reject.

**Details Of Ethics Concerns:**

The fairness concerns arises from lack of clear distinction between the proposed work and the cited papers of SEED and EASE, methods that are integral in the proposed approach (outlined in weakness section). I have also flag the plagiarism checkbox since there is a significant  significant textual overlap between the present submission and SEED and EASE papers (textual similarities are highlighted in the weakness section), that warrant closer attention.

---

### Official Review · Reviewer_JcUB · 2024-10-25

**Soundness:** 2
**Presentation:** 2
**Contribution:** 2
**Rating:** 3
**Confidence:** 4

**Summary:**

The paper introduces ProCEED, a method designed to address challenges in exemplar-free class-incremental learning (EFCIL). The primary contribution of ProCEED is its approach to realigning representations of previous classes in evolving subspaces, thereby reducing the catastrophic forgetting common in EFCIL scenarios. The method leverages a mixture-of-experts (MoE) architecture, Gaussian prototypes, and angular drift compensation to preserve the decision boundaries of past tasks.

**Strengths:**

1. ProCEED combines prototype consolidation and ensemble-based learning to tackle the forgetting problem without relying on previous task data.
2. The experiments are fairly comprehensive, demonstrating ProCEED’s superiority in terms of accuracy.
3. The topic of exemplar-free class-incremental learning is somewhat significant in this era.

**Weaknesses:**

1. The paper lacks novelty, as it appears to be an ensemble of SEED [1] and EASE [2]. Moreover, Section 3.3 is very similar to EASE.
2. The paper does not clearly state the motivation for using MoE, nor does it provide ablation experiments concerning the number of MoE experts. Additionally, it would be beneficial to contrast how ProCEED differs from other MoE-based approaches upfront to contextualize its novelty.
3. It lacks a further discussion on ProCEED’s computational costs and memory demands, which are important, especially for scaling to larger datasets.
4. Given the reliance on MoE and multiple experts, a discussion on the scalability and memory efficiency of ProCEED as tasks increase would enhance the understanding of its practical applicability.
5. Although the paper includes an ablation study (Table 3), it might be helpful to investigate the specific effects of angular drift and logit distillation separately, particularly for each dataset, to better contextualize ProCEED’s performance gains.
6. The experimental results lack reliability because the standard deviation is not being reported.

[1] Divide and not forget: Ensemble of selectively trained experts in Continual Learning. ICLR 2024.

[2] Expandable Subspace Ensemble for Pre-Trained Model-Based Class-Incremental Learning. CVPR2024

**Questions:**

See weaknesses.

---

### Official Review · Reviewer_M5xj · 2024-10-30

**Soundness:** 1
**Presentation:** 1
**Contribution:** 1
**Rating:** 3
**Confidence:** 5

**Summary:**

This paper aims to tackle the exemplar-free class-incremental learning problem. The topic is important to the machine learning field and worth exploring. The authors propose prototype consolidation and expert ensemble to tackle this problem. The proposed method is evaluated on several datasets against other baselines.

**Strengths:**

1. This paper aims to tackle the exemplar-free class-incremental learning problem. The topic is important to the machine learning field and worth exploring.
2. The authors propose prototype consolidation and expert ensemble to tackle this problem.
3. The proposed method is evaluated on several datasets against other baselines.

**Weaknesses:**

1. This paper seems to have extensive overlapping with existing works like [1] and [2]. For example, the expert ensemble process seems to be the same as [1]. The so-called “angular drift compensation via semantic mapping” is the direct copy of [2]. Besides, I also noticed that even the related work part on “Gaussian Models in CL” is also a direct copy of [1]. Figure 2 in this paper is also very similar to [2].
2. The extensive overlapping makes the contribution of this paper doubtful. The manuscript seems like a combination of [1] and [2]. The only difference is that the authors apply the logit-level distillation loss, which is also a common strategy in the class-incremental learning field.
3. Although the experimental results seem to achieve a new state-of-the-art, the major concern lies in the half-baked ablation and comparison protocol. For example, it seems the proposed method has a much higher starting accuracy than other compared methods in Figures 5b and 6a. The authors also introduce an unfair comparison in Table 6, where ProCEED uses task-incremental learning while other compared methods are evaluated in a class-incremental learning scenario. This makes the experimental results doubtful.

[1] DIVIDE AND NOT FORGET: ENSEMBLE OF SELECTIVELY TRAINED EXPERTS IN CONTINUAL LEARNING. ICLR 2024

[2] Expandable Subspace Ensemble for Pre-Trained Model-Based Class-Incremental Learning. CVPR 2024

In conclusion, while the paper presents a modification of existing works, it is important to acknowledge and respect the original contributions. I would recommend the authors to ensure proper attribution and respect for the works they build upon when resubmitting this manuscript.

**Questions:**

See Weaknesses

**Details Of Ethics Concerns:**

This paper seems to have extensive overlapping with existing works like [1] and [2]. For example, the expert ensemble process seems to be the same as [1]. The so-called “angular drift compensation via semantic mapping” is the direct copy of [2]. Besides, I also noticed that even the related work part on “Gaussian Models in CL” is also a direct copy of [1]. Figure 2 in this paper is also very similar to [2].

[1] DIVIDE AND NOT FORGET: ENSEMBLE OF SELECTIVELY TRAINED EXPERTS IN CONTINUAL LEARNING. ICLR 2024

[2] Expandable Subspace Ensemble for Pre-Trained Model-Based Class-Incremental Learning. CVPR 2024

---

### Official Review · Reviewer_iWhm · 2024-10-31

**Soundness:** 3
**Presentation:** 1
**Contribution:** 2
**Rating:** 5
**Confidence:** 4

**Summary:**

This paper presents a method for incremental learning that comprises two key components: (1) multiple experts and (2) prototype compensation. The experimental results demonstrate substantial improvements compared to the baseline models.

**Strengths:**

The experimental results validate most of the author's claims, demonstrating significant performance improvements.

**Weaknesses:**

1) Writing needs improvement. The article lacks a detailed description of the methods, making it difficult to clearly understand the construction details of ProCEED, and some symbols are not clearly explained. For example, the first appearance of \mathcal{F}_{\hat{k},t}, \mathcal{F}_{k,t} and \mathcal{F}^*_{k,t} in Eq(2) is not followed by an explanation, and the relationship between \mathcal{F}_{k,t} and \mathcal{F} is unclear. Readers unfamiliar with Rypes ́c ́ et al. (2023) may find it difficult to grasp the authors’ description of the model.
2) Limited novelty. Though significant performance improvements, ProCEED shows limited novelty as simply combining the approaches of  Rypes ́c ́ et al. (2023) (multiple experts) with Zhou et al. (2024b) (prototype compensation). There is no fundamental innovation in the respective components compared to previous work.
In addition, there are some same  sentences as in previous literature. e.g., line175 "ProCEED comprises K deep network experts F ◦ Gk (here K = 2), sharing the initial layers F for higher computational performance. F are frozen after the first task. Each expert contains one Gaussian distribution per class c ∈ C in his unique latent space." vs Rypes ́c ́ et al. (2023) "SEED comprises K deep network experts g k◦ f (here K = 2), sharing the initial layers f for higher computational performance. f are frozen after the first task. Each expert contains one Gaussian distribution per class c ∈ C in his unique latent space."; line 270 "Without loss of generality, we formulate the above misalignment issue such that given all prototypes (old o and new n) associated with any expert, the target is to project the old class prototypes onto the latest subspace to obtain a new realigned prototype μˆn,o using μo,o, μn,o, and μn,n." vs Zhou et al. (2024b) "Without loss of generality, we formulate the above problem as: given two subspaces (old and new) and two class sets (old and new), the target is to estimate old class prototypes in the new subspace P o,n using P o,o , P n,o , P n,n ."

**Questions:**

1) Eq(4), why can have x from all tasks at the same time to build fk?
2) Line 248, why can have access to the drifted prototypes μ2,1? In my understanding, first train D1 to get F1 and save μ1,1, then train D2 to get F2 and save μ2,2. How can D1 be used to  get μ2,1 when F2 has not yet been built? As shown in Eq(6), the elements above the diagonal are all zero.
3) It seems somewhat unusual to use ResNet32 for the ImageNetSubset; ResNet18 is a more commonly adopted choice.

---

### Note · Authors · 2024-11-12

**Comment:**

Dear Reviewer/Editors,
Thank you for your feedback and the opportunity to address the concerns raised
regarding our paper. We understand the importance of maintaining the integrity of
academic contributions, and we take any comments on plagiarism very seriously.
To clarify, our paper indeed builds upon the foundational ideas introduced in two
prior works, SEED [1] and EASE [2], which have been duly cited in both the
Introduction, Related Works and Experiments sections. From the onset, we have
been transparent about our intellectual inheritance from these studies, and our
paper explicitly acknowledges the significance of these prior contributions to our
research.
Our study extends the work of SEED and EASE by identifying a key limitation within
SEED’s approach—specifically, a potential drift issue with the prototypes when
learning   from   sequential   tasks.   Recognizing   this   challenge,   we   innovatively
addressed it by incorporating cosine similarity as a corrective measure, which was
initially proposed in the EASE paper for pre-trained transformer-based modes. This
specific   methodological   enhancement   and   unique   integration   of   the   two
methodologies   is   our   unique   contribution,   enabling   more   reliable   prototype
alignment in scenarios where PTM-based models are not feasible or applicable.
We have made every effort to delineate our original contributions clearly, and we will
make additional revisions to further emphasize the distinctions between our work
and the referenced studies. Our aim has always been to contribute new insights
while duly respecting and crediting prior research.

Thank you for your time and consideration in reviewing our response. We are
committed to upholding the highest standards of research ethics and remain open to
any additional suggestions that may help clarify our contributions.

Best regards,


[1] DIVIDE AND NOT FORGET: ENSEMBLE OF SELECTIVELY TRAINED EXPERTS IN CONTINUAL LEARNING. ICLR 2024

[2] Expandable Subspace Ensemble for Pre-Trained Model-Based Class-Incremental Learning. CVPR 2024

**Withdrawal Confirmation:**

I have read and agree with the venue's withdrawal policy on behalf of myself and my co-authors.